

# Nitrogen addition enhances seed yield by improving soil enzyme activity and nutrients

Wenbo Mi, Feng Luo, Wenhui Liu, Yan Qin, Yongchao Zhang, Kaiqiang Liu and Wen Li

Key Laboratory of Superior Forage Germplasm in the Qinghai-Tibetan Plateau, Qinghai Academy of Animal Husbandry and Veterinary Sciences, Qinghai University, Xining, China

## ABSTRACT

Nitrogen (N) addition is a simple and effective field management approach to enhancing plant productivity. Nonetheless, the regulatory mechanisms governing nitrogen concentrations and their effect on soil enzyme activity, nutrient levels, and seed yield in the *Festuca kirilowii* seed field have yet to be elucidated. Therefore, this study sought to investigate the effect of N fertilizer application on soil enzyme activities, soil nutrients, and seed yield of *F. kirilowii* Steud cv. Huanhu, the only domesticated variety in the *Festuca* genus of the Poaceae family, was investigated based on two-year field experiments in the Qinghai–Tibet Plateau (QTP). Results showed that N input significantly affected soil nutrients (potential of hydrogen, total nitrogen, organic matter, and total phosphorus). In addition, soil enzyme activities (urease, catalase, sucrase, and nitrate reductase) significantly increased in response to varying N concentrations, inducing changes in soil nutrient contents. Introducing N improved both seed yield and yield components (number of tillers and number of fertile tillers). These findings suggest that the introduction of different concentrations of N fertilizers can stimulate soil enzyme activity, thus hastening nutrient conversion and increasing seed yield. The exhaustive evaluation of the membership function showed that the optimal N fertilizer treatment was N4 (75 kg·hm$^{-2}$) for both 2022 and 2023. This finding provides a practical recommendation for improving the seed production of *F. kirilowii* in QTP.

## INTRODUCTION

*Festuca kirilowii* is one of the scarce grass species capable of large-scale seed production, which plays a key role in ecological restoration and mitigating the challenges confronted by animal husbandry in the Qinghai–Tibet Plateau (QTP). Nonetheless, owing to the low productivity of native seed fields, inefficiencies in eco-restoration, and the pressing need for seed utilization in ecological restoration, the demand for a more abundant supply of *F. kirilowii* seeds arises. Numerous studies have been conducted on nitrogen (N) as a principal limiting factor in the growth and development of most plants (*Amanullah et al., 2016*; *Wang et al., 2018b*). For perennial forage grass, nitrogen fertilization increases seed

Corresponding author
Wenhui Liu, qhliuwenhui@163.com

yield across different plant ages (*Zhang, Li & Li, 2001*; *Li et al., 2012*; *Pei-Sheng et al., 2001*). Accordingly, N application is widely used in seed production and stands as one of the key factors influencing both seed quality and yield. Inadequate N addition leads to reduced seed yield owing to reduced returns for cultivators. However, excessive N addition fails to produce a substantial increase in improved seed yield, owing to the principle of diminishing returns, and results in escalated costs (*Cassman, Dobermann & Walters, 2003*; *Kuai et al., 2023*; *Reussi Calvo et al., 2013*). In addition, the optimal amount of N to be applied varies depending on species, soil fertility, planting density, and climatic conditions (*Chen et al., 2023*; *Peng et al., 2010*; *Yang et al., 2019*). Previous studies on *Elymus nutans* have indicated that an optimal N addition rate of 250 kg·hm² can enhance seed yield (*Guo-Ping et al., 2010*). *Yuan et al.*'s *(2022)* study on *Kengyilia melanthera*, N fertilizer application ranging from 0 to 240 kg·hm$^{-2}$ demonstrated that the seed yield and yield components (number of tillers, number of fertile tillers) reached their peak values under 180 kg·hm$^{-2}$. Therefore, studying the optimal nitrogen application concentration for *F. kirilowii* seed yield is crucial.

Soil is a complex microenvironment, and its nutrients are regulated by various factors (*Wang et al., 2023b*; *Zhao et al., 2022*), including soil enzymes, physical and chemical properties, and microorganisms. N fertilizer directly affects soil nutrient balance, affecting soil organic matter (SOM) storage in grassland ecosystems (*Chen et al., 2020*; *Li, Yan & Zhang, 2023a*; *Liao et al., 2021*). N application induces changes in soil pH, influencing soil microbial activity and major soil biochemical reactions (*Gong et al., 2015*; *Song et al., 2014*; *Yang et al., 2023a*). Soil enzymes originate from cellular secretions of animals, plants, and microorganisms in the soil and the breakdown products of their residues (*Davies, Coulter & Pagliari, 2022*). These enzymes accumulate in the soil as free enzymes or are adsorbed on the surface of soil particles; moreover, they can reflect soil carbon and nitrogen transformations (*Rao et al., 2014*). Enzymes commonly used to reflect soil carbon and nitrogen transformations can be categorized into oxidoreductases and hydrolases. Soil oxidoreductases primarily include catalase (CAT), nitrate reductase (NR), and hydroxylamine reductase (HR), which are directly involved in the denitrification process of soil nitrogen. Soil hydrolases, such as urease (UE) and sucrase (SC), hydrolyze proteins and other macromolecules into simple small molecules that are easily absorbed and used by plants. Soil enzyme activities are sensitive to the amount of N applied. N application may indirectly influence soil enzyme activities by altering soil physicochemical properties (*Li et al., 2023b*; *Nannipieri et al., 2012*). Previous research has revealed various responses of soil enzyme activity to N addition (*Song et al., 2009*; *Zhang et al., 2023a*). In some studies, N addition significantly increased UE activity due to increased microbial demand for C, N, and P; by contrast, others reported a negative effect on UE activity (*Saiya-Cork, Sinsabaugh & Zak, 2002*; *Stone et al., 2012*), particularly during the early growing season (*Andersson, Kj?ller & Struwe, 2004*; *Weintraub & Schimel, 2005*; *Zhang et al., 2022*). The effect of N on soil enzyme activity has been attributed to variations in (i) nutrient effectiveness, (ii) litter and soil C:N ratios, and (iii) microbial biomass (*Treseder & Vitousek, 2001*; *Ullah et al., 2023*; *Xu et al., 2022*). Increasing N effectiveness affects soil

enzyme function and activity, promoting SOM decomposition and stimulating plant growth by alleviating N limitation (*Cusack & Daniela, 2013*; *Qin et al., 2023*).

However, existing research has focused mainly on annual field crops, with fewer studies conducted on perennial forages. Relevant studies are limited to individual studies on the effects of N application solely on seed yield, soil nutrients, and soil enzyme activities (*Wang et al., 2018a*; *Yuan et al., 2022*; *Zhao et al., 2022*). In the current study, we measured the seed yield, yield components, soil physicochemical, and soil enzyme activities after N addition to elucidate the process by which the interaction of these factors influences seed yield. This study not only addresses the lack of research on perennial forage grasses in this context but also suggests potential solutions for ecological management and the development of animal husbandry in QTP.

# MATERIALS AND METHODS

## Experimental sites

The experiment was conducted for two consecutive years, from 2022 to 2023, at the national perennial forage germplasm resource nursery of Xihai Town, Haibei Prefecture, Qinghai Province, China (36°59′36″N, 100°52′85″E, 3,156 m a.s.l.). The climate is cold, with no absolute frost-free period. The annual average temperature is 0.5 °C, and annual precipitation is 369.1 mm, mostly concentrated between July and September. Annual evaporation measures 1,400 mm, with an annual sunshine duration of 2,980 h, and an average frost-free period of about 93 d. The soil of the experimental land is classified as chestnut soil, and its chemical properties are listed in Table S1. Mean monthly temperatures and rainfall for 2022 and 2023 are presented in Fig. S1. Abbreviations are listed in Table S2.

## Experimental management

Seeds of *F. kirilowii* (specifically the variety *Festuca kirilowii Steud* cv. Huanhu) were supplied by the Grassland Institute of Qinghai Academy of Animal Science and Veterinary Science. The seeds had a germination rate of 81% and a 1,000-seed weight of 0.622 g.

## Experimental design

Depending on the local climate and sowing dates, the trial sites were sown in early July 2021 and July 2022. Before sowing, calcium superphosphate (150 kg·hm$^{-2}$) was applied as a basal fertilizer. The sowing rate was 22.5 kg·hm$^{-2}$, with a row spacing of 50 cm and a seeding depth of 3–5 cm. Fertilizer treatments (urea fertilizer, 46% N) were applied to the experimental plots during the regreening stage in mid-June of 2022 and 2023. The trial employed a completely randomized block group design, comprising seven nitrogen application treatments (N1, 0; N2, 25; N3, 50; N4, 75; N5, 100; N6, 125; and N7, 150 kg·hm$^{-2}$). Each N fertilizer treatment had three replication plots, resulting in a total of 21 plots. Experimental units consisted of plots with an area of 20 m$^2$ (5 m × 4 m) and eight rows. Seeds were not sown within 50 cm of the plot boundary to reduce the marginal effect. Hand weeding was performed twice during the reproductive period. No irrigation was conducted, and no pesticides were applied, given the low incidence of pests and diseases.

## Yield components and seed yield

The number of tillers (NTs) and the number of fertile tillers (NFTs) were measured to assess yield. During the full-blooming stage, NTs and NFTs were measured in a randomly selected row from each sample plot, and three replicates were obtained from each sample plot.

Approximately 25 days after the peak of flowering, all seeds in each plot were harvested, except those for the side rows. The harvested seeds were threshed, cleaned, and naturally dried to determine the seed yield (SY) per hectare.

## Soil enzyme activities

We collected soil samples from three randomly selected locations within each plot during the 2022 and 2023 seed harvests. Soil samples were extracted from the 0–20 and 20–40 cm layer by using a cylindrical soil sampler. Each soil sample composite was homogenized, passed through a 2 mm sieve, and divided into two parts. A portion of the soil sample was stored at 4 °C for subsequent content analysis of soil enzyme activity. Another portion was air-dried before soil physical and chemical analysis. Soil enzyme (UE, CAT, SC, and NR) activities were determined using an enzyme activity detection kit (soil urease kit, soil catalase kit, soil sucrase kit, and soil NR kit) (Suzhou Mengxi Biomedical Technology Co., Ltd.). The enzyme urease used urea as the substrate (*Elsas, 1995*), CAT used $H_2O_2$ (*Saiya-Cork, Sinsabaugh & Zak, 2002*), SC used sucrose (*Yang et al., 2023b*), and NR used $KNO_3$ (*Nardi et al., 2005*). The detailed assay procedures for these soil enzyme activities are provided in the kit program.

## Soil nutrients

Soil pH was determined using the digital pH meter (Seven2Go, Mettler-Toledo Instruments (Shanghai) Co., Ltd, Shanghai, China) in a 1:5 soil/water suspension after shaking for 5 min. Soil physicochemical properties were determined in accordance with the guidelines outlined in "Soil Sampling and Methods of Analysis (*Crepin & Johnson, 1993*)." Soil organic matter (SOM) was determined using the $K_2Cr_2O_7$ redox titration method (*Maxwell et al., 2023*). Total nitrogen (TN) in the soil was assessed *via* Kjeldahl digestion (*Keeney, 1982*). The steps are as follows: The samples were pretreated, followed by the sequential addition of $HgCl_2$ solution, $NH_4Cl$ solution, and KI solution. The mixture was then filtered and measured using a spectrophotometer. Total phosphorus (TP) content was measured *via* antimony–molybdenum colorimetry after perchloric acid digestion (*Wezel, Rajot & Herbrig, 2000*). This step involved accurately weighing 0.5 to 1 g of air-dried soil samples that passed through a 100 mesh sieve. The samples were placed in a 100 mL digestion tube, to which 8 mL of concentrated $H_2SO_4$ and 10 drops of 70–72% HClO were added. The mixture was then decocted for 40–60 min. The cooled decocted solution was poured into a 100 mL volumetric flask, and water was added to adjust the volume. A volume of 5 mL of the clarified solution was aspirated, and two drops of dinitrophenol indicator were added. Subsequently, 4 mol·L$^{-1}$ of NaOH solution was slowly added dropwise until the solution turned yellow; 2 mol·L$^{-1}$ (½$H_2SO_4$) was added with one drop, ensuring the fading of the yellow color of the solution. Approximately 5 mL of

molybdenum antimony reagent was incorporated, and water was added to reach a total volume of 50 mL. The mixture was shaken thoroughly and then allowed to sit for 30 min. Colorimetry was performed at a wavelength of 880 mm.

## Statistical analysis

The effect of N addition on soil properties, enzyme activities, seed yield, and yield components was investigated by one-way analysis of variance (ANOVA) using IBM Statistical Package, SPSS v. 27.0 (IBM, Armonk, NY, USA). A two-way ANOVA with a general linear model at $P < 0.05$ was applied to determine the combined effect of N fertilizer and trial years on seed yield and yield components. Two-way ANOVA was also employed to assess the combined effect of N fertilizer and soil layers on soil properties and enzyme activities. Correlation and path analysis were conducted using SPSS v. 25.0 to evaluate the effects of soil factors and yield components on seed yield. Graphs were generated using Origin 2021 (OriginLab, Massachusetts) and R 4.3.1 (*R Development Core Team, 2023*) software. The R code is provided in the Supplementary Material.

The membership function method was used to investigate the response of yield components, seed yield, soil nutrients, and soil enzyme activities to N fertilizer, as well as to determine the optimal amount of N to apply for *Festuca kirilowii Steud* cv. Huanhu. The calculation formula for the forward membership function is expressed as $y = (x_a - x_{min})/(x_{max} - x_{min})$, and that for the negative membership function is $y = 1-(x_a - x_{min})/(x_{max} - x_{min})$. In these formulas, $x_a$ is the value of a specific index, and $x_{max}$ and $x_{min}$ denote the maximum and minimum values of an indicator under different N application treatments.

# RESULT

## Soil nutrients

The ANOVA results for soil nutrients under different treatments are summarized in Table 1. N addition and the soil layer significantly affected pH, SOM, TN, and TP ($P < 0.05$). However, the interaction between N addition and soil layer showed non-significant effects for the aforementioned soil nutrients ($P > 0.05$).

The specific results of changes in soil nutrients under different treatments are depicted in Fig. 1. The negative effect of N addition on soil pH decreased with increasing nitrogen concentrations. However, SOM, TN, and TP exhibited an increasing and then decreasing trend with increasing N. SOM, TN, and TP reached maximum contents under N3, N4, and N4 conditions, respectively, significantly exceeding the content under N1 conditions. No significant differences in soil pH and SOM were found between the two soil layers under similar treatment conditions. TN was significantly higher in the 0–20 cm soil layer than in the 20–40 cm soil layer. Under N4–N7 treatment conditions, TP content was significantly higher in the 0–20 cm than the 20–40 cm soil layer.

## Soil enzyme activities

The ANOVA results for each enzyme under each treatment are listed in Table 2. N addition and soil layer exerted significant effects on four soil enzyme activities ($P < 0.05$).

**Table 1 Analysis of variance of the effects of nitrogen fertilizer and soil layers on *F. kirilowii* soil nutrients.**

| Index | Source of variance | Df | F-ratio | P-value |
|---|---|---|---|---|
| PH | Soil (S) | 1 | 10.517 | 0.003 |
| | N addition (N) | 6 | 6.500 | 0.000 |
| | Interaction S × N | 6 | 0.291 | 0.936 |
| SOM/g·kg$^{-1}$ | Soil (S) | 1 | 25.378 | 0.000 |
| | N addition (N) | 6 | 2.867 | 0.026 |
| | Interaction S × N | 6 | 0.211 | 0.970 |
| TN/g·kg$^{-1}$ | Soil (S) | 1 | 189.370 | 0.000 |
| | N addition (N) | 6 | 2.366 | 0.045 |
| | Interaction S × N | 6 | 0.350 | 0.904 |
| TP/g·kg$^{-1}$ | Soil (S) | 1 | 42.752 | 0.000 |
| | N addition (N) | 6 | 3.334 | 0.013 |
| | Interaction S × N | 6 | 1.540 | 0.202 |

**Note:**
PH, potential of hydrogen; SOM, soil organic matter; TN, total nitrogen; TP, total phosphorus.

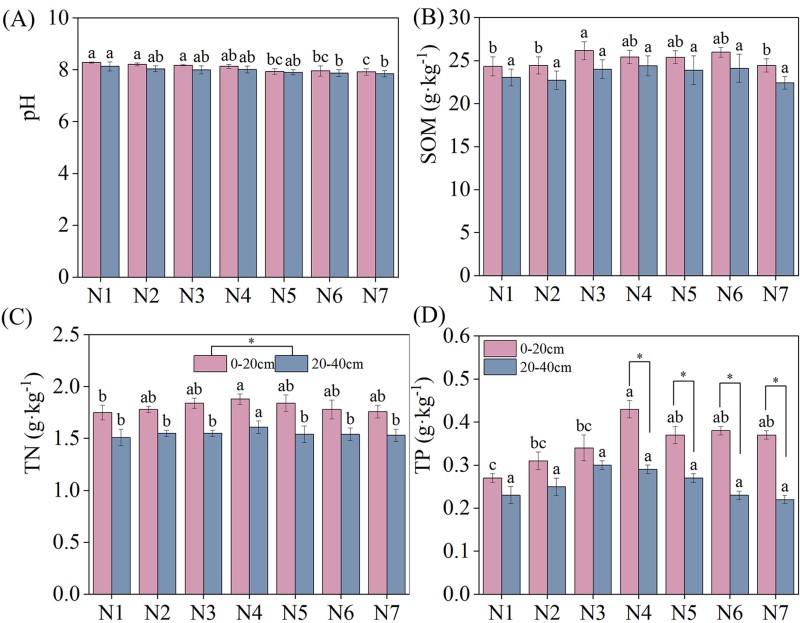

**Figure 1 Effects of nitrogen fertilizer and soil layers treatments on soil nutrients of *F. kirilowii*.** (A) pH, (B) SOM, (C) TN, and (D) TP. Different letters indicate significant differences between different nitrogen fertilizer for the same soil layers treatments groups at the 0.05 level. "*" indicates significant differences between different soil layers for the same N application treatment groups at the 0.05 level. Vertical bar represents the standard error of mean (*n* = 3).

However, the soil enzyme activities exhibited a non-significant effect under the N addition–soil layer interaction (*P* > 0.05).

Soil enzyme activities under different N application treatments are depicted in Fig. 2. Nitrogen application significantly increased soil UE activity (*P* < 0.05). With an increase in

**Table 2 Analysis of variance of the effects of nitrogen fertilizer and soil layers on *F. kirilowii* soil enzyme activities.**

| Index | Source of variance | Df | F-ratio | P-value |
|---|---|---|---|---|
| UE/mg·g$^{-1}$·24 h$^{-1}$ | Soil (S) | 1 | 5.997 | 0.021 |
| | N addition (N) | 6 | 161.212 | 0.000 |
| | Interaction S × N | 6 | 0.184 | 0.979 |
| CAT/mg·g$^{-1}$·24 h$^{-1}$ | Soil (S) | 1 | 14.912 | 0.001 |
| | N addition (N) | 6 | 30.764 | 0.000 |
| | Interaction S × N | 6 | 1.682 | 0.162 |
| SC/mg·g$^{-1}$·24 h$^{-1}$ | Soil (S) | 1 | 6.855 | 0.014 |
| | N addition (N) | 6 | 20.199 | 0.000 |
| | Interaction S × N | 6 | 0.499 | 0.803 |
| NR/mg·g$^{-1}$·24 h$^{-1}$ | Soil (S) | 1 | 17.972 | 0.000 |
| | N addition (N) | 6 | 14.173 | 0.000 |
| | Interaction S × N | 6 | 0.370 | 0.892 |

**Note:**
UE, urease; CAT, catalase; SC, sucrase; NR, nitrate reductase.

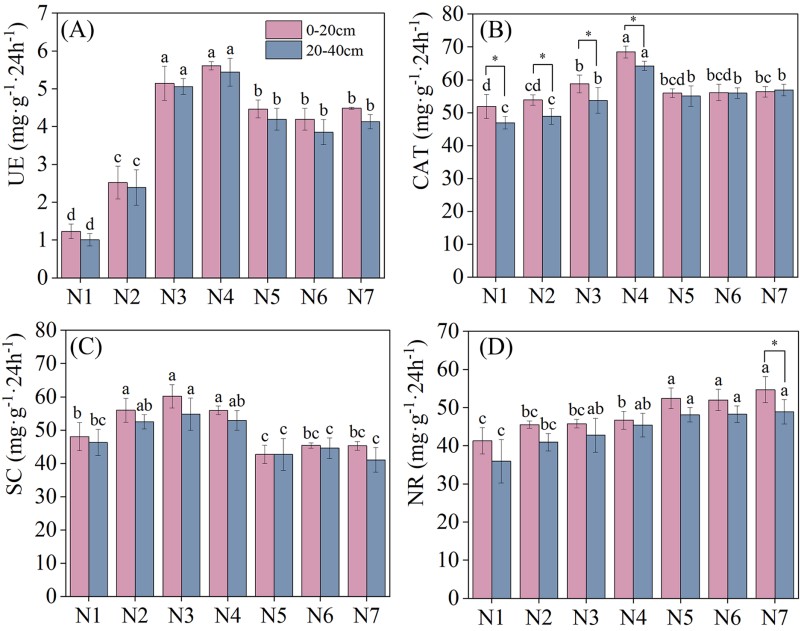

**Figure 2 Effects of nitrogen fertilizer and soil layers treatments on soil enzymes activity of *F. kirilowii*.** (A) UE, (B) CAT, (C) SC, and (D) NR. Different letters indicate significant differences between different nitrogen fertilizer for the same soil layers treatments groups at the 0.05 level. "*" indicates significant differences between different soil layers for the same N application treatment groups at the 0.05 level. Vertical bar represents the standard error of mean (*n* = 3).

nitrogen application, soil UE activity in different soil layers exhibited a pattern of initial increase, followed by a subsequent decrease, with all reaching maximum urease activity under N4 treatment. Under N4 treatment, soil urease activity increased by 356.10% and 449.49% in the 0–20 cm and 20–40 cm soil layers, respectively, relative to that in N1.

Meanwhile, compared with 0–20 cm soil layer, soil UE activity was lower in the 20–40 cm soil layer than that in the 0–20 cm soil later under all nitrogen application treatments. CAT and SC activities reached their maximum levels under N4 and N3 treatments, respectively. No significant difference in SC activity across different soil layers was found under nitrogen application treatment. Notably, CAT activity was significantly higher in the 0–20 cm than in the 20–40 cm soil layer under N1–N4 treatments. NR activity increased as nitrogen concentrations were increased, indicating a positive effect of N addition. No significant differences in NR activity were found between the two soil layers under all treatments except N7.

## Yield components and seed yield

The ANOVA results for yield components and SY under various treatments are listed in Table 3. N addition exerted significant effects on NTs, NFTs, and SY ($P < 0.01$). No significant differences were observed in all metrics measured in 2022 and 2023.

The NTs and NFTs in both trial years were significantly higher under the N2–N6 treatment than under N1 treatment ($P < 0.05$, Fig. 3). Their peak values were observed under N3 or N4. With increasing rates of N fertilizer application, the seed yield showed a pattern of initial increase followed by a subsequent decrease (Fig. 4), reaching their peak values under N4 treatment in both 2022 and 2023. Notably, excessive application of N fertilizer (N6 or N7) exhibited a non-significant increase in seed yield ($P > 0.05$). This increase was accompanied by a reduction in two seed yield component indicators (Fig. 3). Seed yield exhibited fluctuations during the 2-year independent trials in 2022 and 2023 under similar treatment; however, the difference was non-significant ($P > 0.05$).

## Correlation and pathway analysis

The correlations between SY and yield components, soil enzyme activity, and soil physicochemical properties in 2022 and 2023 are presented in Table 4 and Fig. 5. A significant correlation exists between yield components (NTs and NFTs) and SY ($P < 0.01$). SY in 2022 was significantly and positively correlated with UE, CAT, SC, TP, and TN in the 0–20 cm soil layer ($P < 0.05$), and positively correlated with NR, pH, and SOM. SY in 2023 was significantly and positively correlated with UE, CAT, SC, and TN in the 0–20 cm soil layer ($P < 0.05$), and positively correlated with NR, pH, SOM, and TP. In the 20–40 cm soil layer, SY in 2022 was significantly and positively correlated with UE, CAT, SC, and TN ($P < 0.05$). Meanwhile, SY in 2023 was significantly and positively correlated with UE, CAT, SC, TN, and TP ($P < 0.05$).

Subsequently, a path analysis of the metrics (NTs, NFTs, UE, CAT, SC, TN, and TP) that were significantly associated with SY was conducted. In this case, the average values of SY and yield components over 2022 and 2023 were determined, and the average soil enzyme activities and soil nutrients were calculated over two soil layers. The path analysis elucidated 84% of the variation in SY (Fig. 6). SY was directly influenced by soil enzyme activities (UE, CAT, and SC), soil nutrients (TP and TN), and yield components (NTs and NFTs). Soil enzyme activities (UE, CAT, and SC) and soil nutrients (TP and TN) also indirectly affected SY by positively influencing NTs and NFTs. In addition, N addition

**Table 3 Analysis of variance of the influence of N fertilizer on *F. kirilowii* yield components and seed yield.**

| Index | Source of variance | Df | F-ratio | P-value |
|---|---|---|---|---|
| NTs/m$^2$ | Year (Y) | 1 | 0.075 | 0.787 |
| | N addition (N) | 6 | 15.928 | 0.000 |
| | Interaction Y × N | 6 | 0.610 | 0.720 |
| NFTs/m$^2$ | Year (Y) | 1 | 0.000 | 0.984 |
| | N addition (N) | 6 | 109.572 | 0.000 |
| | Interaction Y × N | 6 | 0.892 | 0.514 |
| SY/kg·hm$^{-2}$ | Year (Y) | 1 | 0.020 | 0.887 |
| | N addition (N) | 6 | 23.327 | 0.000 |
| | Interaction Y × N | 6 | 0.176 | 0.981 |

Note:
NTs, number of tillers; NFTs , number of fertile tillers; SY, seed yield.

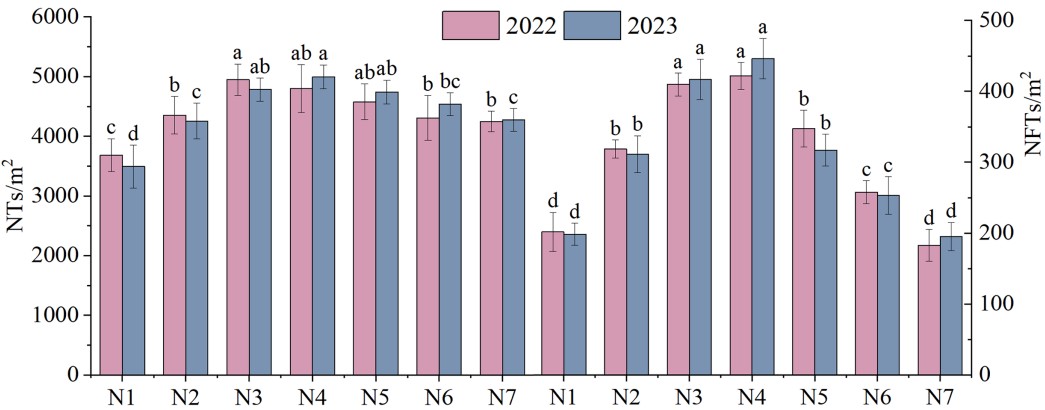

**Figure 3 The yield components of *F. kirilowii* as influenced by seven different rates of N fertilizer application during the trial years 2022–2023.** Different letters indicate significant differences between different nitrogen fertilizer treatments groups at the 0.05 level. Vertical bar represents the standard error of mean.

significantly stimulated soil enzyme activities by increasing soil nutrient content, enhancing the SY of *F. kirilowii*. The standardized effect sizes of the contributing factors further clarify the results of the path analysis (Fig. 7). The findings suggest that NFTs, UE, and TN play significant roles in the formation of SY in *F. kirilowii*.

## Comprehensive analysis of membership function

The changes in the seed trait indices of *F. kirilowii* demonstrated a certain degree of correlation and difference based on the N fertilizer, and a single indicator cannot fully capture the influence of treatment on seed production. Therefore, a comprehensive evaluation of multiple indices was necessary to determine the best combination. The membership function values were calculated based on the values of each indicator under different treatments. These indicators included the indices of four soil enzyme activities,

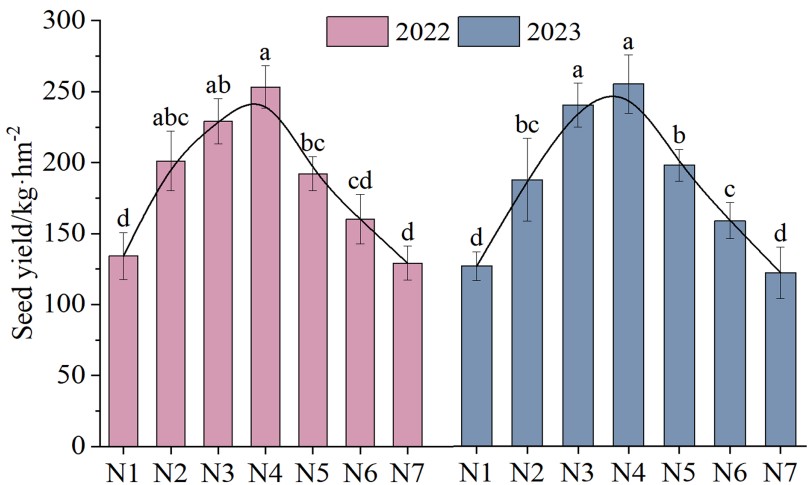

**Figure 4** **The seed yield (mean ± standard error, $n = 4$) of *F. kirilowii* as influenced by N fertilizer during the trial years of 2022–2023.** Different lower-case letters above the column indicate significant differences under different nitrogen fertilizer treatments ($P < 0.05$). Vertical bar represents the standard error of mean.

**Table 4 Correlation between seed yield and yield components during trial year 2022 and 2023.**

| Year | 2022 | | 2023 | |
|---|---|---|---|---|
| Source | NTs | NFTs | NTs | NFTs |
| Seed yield | 0.57** | 0.84** | 0.75** | 0.91** |

Note:
Two asterisks (**) represent significant differences at the 0.01 level.

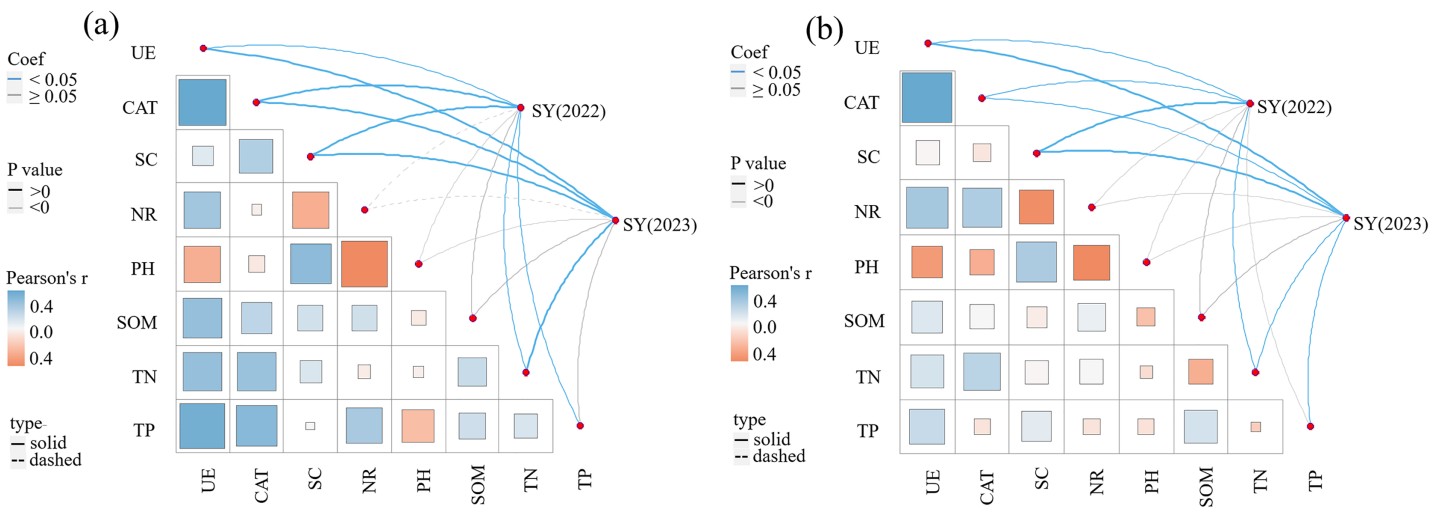

**Figure 5** **Pearson correlation between seed yield and soil enzyme activities, soil nutrients.** (A) 0–20 cm soil layer; (B) 20–40 cm soil layer.

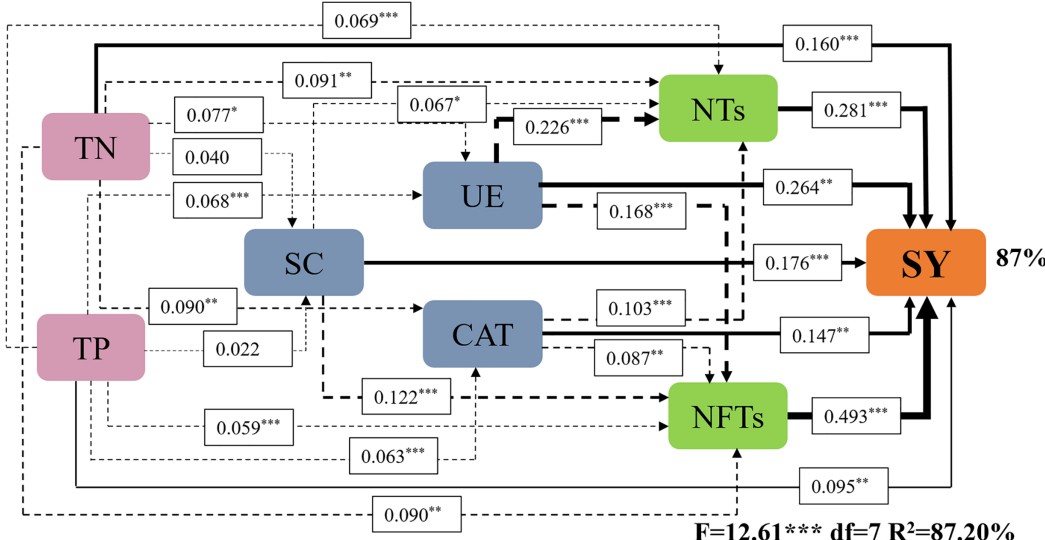

**Figure 6 Effects of soil enzyme activities (UE, CAT, and SC), soil physicochemical properties (TP and TN), and yield components (NTs and NFTs) on seed yield.** TN, Total nitrogen; TP, Total phosphorus; SC, Sucrase; UE, Urease; CAT, Catalase; NTs, Number of tillers; NFTs, Number of fertile tillers; SY, Seed yield. The full and dashed lines represent direct and indirect effects, respectively. The width of lines is proportional to the strength of path coefficients. *, **, and *** represent significant differences at the 0.05, 0.01, and 0.001 level, respectively.

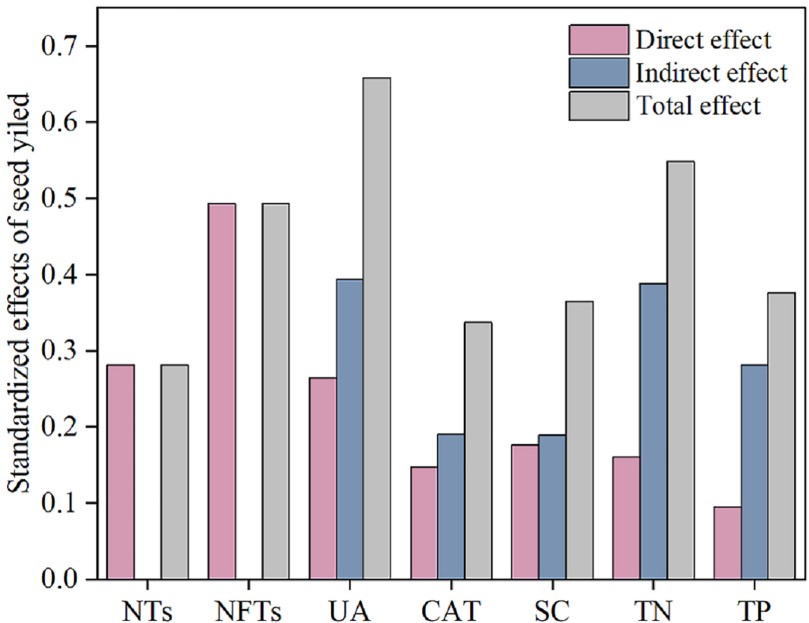

**Figure 7 Standardized effects of indicators on seed yield.**

four soil nutrients, and three yield components (Table 5). Higher membership function values represented a more distinct improvement in *F. kirilowii* seed yield. The final ranking was as follows: N4 > N3 > N5 > N2 > N6 > N7 > N1. The results identified N4 treatment as the most suitable treatment for *F. kirilowii* seed production.

**Table 5 Membership function analysis based on multiple indicators.**

| Indexes | N1 | N2 | N3 | N4 | N5 | N6 | N7 |
|---|---|---|---|---|---|---|---|
| SY | 0.038 | 0.535 | 0.848 | 1.000 | 0.540 | 0.262 | 0.000 |
| NTs | 0.000 | 0.547 | 0.975 | 1.000 | 0.818 | 0.638 | 0.513 |
| NFTs | 0.045 | 0.514 | 0.916 | 1.000 | 0.584 | 0.271 | 0.000 |
| UE | 0.000 | 0.305 | 0.904 | 1.000 | 0.729 | 0.659 | 0.724 |
| CAT | 0.000 | 0.115 | 0.404 | 1.000 | 0.360 | 0.392 | 0.426 |
| SC | 0.302 | 0.781 | 1.000 | 0.793 | 0.000 | 0.154 | 0.032 |
| NR | 1.000 | 0.651 | 0.572 | 0.437 | 0.117 | 0.128 | 0.000 |
| PH | 0.000 | 0.259 | 0.363 | 0.420 | 0.896 | 0.907 | 1.000 |
| SOM | 0.158 | 0.098 | 1.000 | 0.894 | 0.735 | 0.973 | 0.000 |
| TN | 0.000 | 0.324 | 0.618 | 1.000 | 0.559 | 0.279 | 0.132 |
| TP | 0.000 | 0.294 | 0.676 | 1.000 | 0.662 | 0.529 | 0.412 |
| Rank | 7 | 4 | 2 | 1 | 3 | 5 | 6 |

# DISCUSSION

The specific processes and mechanisms of N fertilizer transformation in the plant–soil system are governed by various factors. Soil enzymes, which are important biological activators of energy flow and material cycling in the subsurface, are involved in the transformation of soil nutrients. Alterations in soil enzyme activities can effectively indicate changes in soil physicochemical properties and soil nutrients, thus influencing plant growth regulation. Therefore, investigating the mechanism linking nitrogen-regulated soil enzymes to soil nutrients and their regulatory effect on the formation of plant seed yield needs to be explored.

## Effect of N addition on soil nutrients

The soil nutrient balance is crucial in determining the direction of soil fertility evolution, closely linked to the benefits derived from fertilizer application as well as ecological and environmental safety (*Bindraban et al., 2000*). Historically, the emphasis of fertilizer application has been geared toward increasing yields, ignoring, to a certain extent, issues related to soil fertility balance and ecological and environmental safety (*Saha et al., 2007*). The escalating global deposition of N has led to soil acidification, thus becoming a global environmental problem (*Gong et al., 2015*). The results of the current study align with this trend, revealing a reduction in soil pH. This occurrence may be due to the increase in $NH_4^+$ and $NO_3^-$ contents in the soil by N input; in addition, the replacement of $NH_4^+$ with base cations ($Ca^{2+}$, $Mg^{2+}$, $K^+$, $Na^+$) on the surface of the soil colloid renders the base cations susceptible to leaching. When $NH_4^+$ was absorbed by plants, it released $H^+$ into the soil solution. Concurrently, nitrogen promotes cation uptake by plant roots, leading to a substantial release of $H^+$ and ultimately, a decrease in soil pH (*Alatalo et al., 2017*; *Zhang et al., 2023b*). In the present study, N application enhanced the soil nutrients (SOM, TN, and TP) content, particularly increasing the TP content. This effect may be attributed to the coordinating effect between nitrogen and phosphorus, which promotes phosphorus
with nitrogen. The maximum level of SOM was observed under the N3 treatment, significantly exceeding that in the control (N1); no significant difference in SOM was found between the two soil layers. The TN and TP contents reached their maximum values under N4 treatment. An appropriate amount of N can improve the soil nutrient content because it can strengthen microbial metabolism in the soil (*Gong et al., 2015*; *Presley et al., 2012*). In addition, the subsequent increase in secreted metabolic products facilitates the mineralization of humus and biodegradation-resistant organic matter in the soil, thus improving overall soil nutrient content. Conversely, the excessive application of N and the stacking effect of N fertilizers can induce the accumulation of excessive N in the soil, disrupting the nutrient balance within the soil ecosystem and compromising the biological attributes of the soil. These disturbances can have a detrimental effect on the ability of the soil to be used sustainably (*Jing et al., 2007*; *Bimüller et al., 2014*). In addition, the TN and TP contents in the 0–20 cm soil layer are significantly higher than those in the 20–40 cm soil layer. This disparity can be attributed to the reduced rainfall in the QTP, resulting in reduced nutrient leaching into the deeper soil layers (*Chu et al., 2021*).

## Effect of N addition on soil enzyme activities

Soil enzymes serve as important indicators of soil fertility, with their activity closely related to soil nutrients. They reflect the strength of soil nutrient transformation, characterizing soil fertility (*Yang et al., 2012*). UE, a significant hydrolase, affects the soil N cycle. Specifically, it converts limited amounts of nitrogenous organic substrates into inorganic nitrogenous compounds (*e.g.*, ammonia) to provide available nitrogen for normal plant growth and development (*Iovieno et al., 2009*). As an important hydrolyzing enzyme in the soil, SC can degrade and convert sucrose in the soil into glucose and fructose, providing carbon and energy supply for microbial activities (*Weige et al., 2016*). N addition has been shown to significantly increase soil UE and SC activities (*Gong et al., 2015*; *Weng et al., 2013*). Consistent with this finding, the present study indicates that N application significantly increased UE and SC activities in both surface (0–20 cm) and deep soils (20–40 cm). The difference was that UE activity reached its maximum under the N4 treatment, whereas SC activity was highest under the N3 treatment. The enhanced UE and SC activities after under N application may be attributed to the stimulation of microbial activity. This stimulation arises from the increased availability of soil N and the enhancement of soil physicochemical properties, rendering the soil environment more suitable for microbial growth and proliferation (*Iovieno et al., 2009*; *Nayak, Babu & Adhya, 2007*). Second, UE is mainly derived from root secretion, and the introduction of moderate nitrogen favors root growth, leading to increased root secretion and thus heightened UE activity (*Polacco, 1977*). As key oxidoreductases in soil, CAT and NR play important roles in energy transfer and material metabolism of organisms in soil. They also indirectly affect plant uptake and utilization of N (*Pu et al., 2019*). NR catalyzes $NO_3^-$ and $NO_2^-$ and then transform them into $NO_2^-$ and NO, respectively (*Braker & Conrad, 2011*). Nitrification, denitrification, and mineralization induced by these enzymes are important biochemical processes in soil N metabolism that affect N use and loss. Therefore, various factors affecting soil N metabolism, particularly N addition, are also regarded as major

determinants of these enzyme activities (*Liu et al., 2018*). Exogenous N directly affects soil properties and microbial composition, indirectly increasing soil NR and CAT activities (*Wang et al., 2023a*; *Zhao et al., 2022*). In our experiment, CAT activity increased first and then decreased as N application increased, and CAT activity exceeded those in the control under all treatments. By contrast, NR activity positively correlated with N application. This trend may be due attributed to the promotion of crop root metabolism by N fertilizer, leading to heightened root secretion and accelerated microbial reproduction. By absorbing nutrients from the soil, inter-root microorganisms can form near-root, slow-acting nutrient pools, contributing to the enhancement of soil enzyme activities (*Carreiro et al., 2000*; *Weng et al., 2013*). In addition, the current study found that N application exerted no significant effect on NR activity in the 0–20 and 20–40 cm soil layers. Enzyme activities decreased with the deepening of the soil layer. This may be due to the fact that soil microorganisms are mainly active in the tillage layer where the root system is well developed.

### Effect of N addition on seed yield and yield components

N is an essential nutrient for plant growth and development, and proper fertilization management positively affects crop growth, yield formation, and ecological and environmental protection (*Duan et al., 2011*; *Feng et al., 2018*). For perennial herbage, the application of appropriate quantities of N can improve aboveground biomass yield and grassland stand vegetation coverage, as well as meet the demands of grassland animal husbandry and ecological restoration. In addition, this treatment can effectively increase seed yield, as well as prolong the lifespan of forage seed fields (*Gislum & Boelt, 2009*; *Zhao et al., 2023*). Nonetheless, seed yield constantly relies on the multiplicative interaction of various yield components (NTs, NFTs, and TSW) (*Dewitt et al., 2021*). Numerous studies have demonstrated that appropriate nitrogen supplementation significantly enhances the seed yield components. Research on *Elymus sibiricus* (*Wang et al., 2018b*), *Elymus nutans* (*Guo-Ping et al., 2010*), and *Poa pratensis* (*Bai et al., 2010*) has demonstrated that N fertilization improves seed yield by increasing NTs and (or) NFTs. Investigations on the seed yield of tall fescue showed a positive correlation between N fertilizer application and NFTs within the 60–120 kg·hm$^{-2}$ (*Hui et al., 2003*). Similar findings have been reported on seed yield in *Lolium perenne*. N addition increased NTs, NFTs, and TSW (*Hides, Kute & Marshall, 2010*). In the current study, N application significantly increased NTs, NFTs, and seed yield. NTs (r = 0.75, $P < 0.01$) and NFTs (r = 0.91, $P < 0.01$) exhibited a high positive correlation with seed yield with N fertilizer application in the 2023 study. Path analysis showed that NTs and NFTs significantly contributed to seed yield. These results demonstrate that N addition was one of the most effective approaches to improving *F. kirilowii* seed yield by increasing NTs and NFTs.

### Correlation and path analysis

Soil enzyme activity, physicochemical properties, and seed yield components directly or indirectly affect seed yield formation. Nitrogen fertilization serves as a highly effective cultivation measure for increasing seed yield (*Weng et al., 2013*). Appropriate nitrogen

addition can markedly enhance soil nutrient content, particularly TN and TP. This occurrence thus stimulates soil enzyme activities (UE, CAT, and SC), inducing heightened soil enzyme activities. Consequently, factors contributing to seed yield—NTs and NFTs—increase, ultimately improving seed yield. Path analysis identified NFTs, UE, and TN as the most effective indicators for seed yield in *F. kirilowii*. The total effects of NFTs, UE, and TN on seed yield were 0.493, 0.658, and 0.548, respectively. Membership function analysis showed that *F. kirilowii* demonstrated the highest ranking under the N4 (75 kg·hm$^{-2}$) treatment, indicating that the application of 75 kg·hm$^{-2}$ N fertilizer was optimal for seed yield formation. This finding provides insights into the cultivation and promotion of high-quality forages in QTP. Whereas this study was limited to exploring the effect of N fertilization on seed yield, future research could explore the mixed application of multiple fertilizers (*e.g.*, nitrogen fertilizer + phosphorus fertilizer, organic fertilizer + inorganic fertilizer) to obtain higher seed yield and more environmentally friendly cultivation practices than usual.

## CONCLUSION

This study found that N addition significantly affected soil nutrient content, soil enzyme activity, yield components, and seed yield. Notably, the maximum values of TN, TP, UE, CAT, NFTs, and SY were reached under the N4 treatment. Soil enzyme activities (UE, CAT, and NR) under each N application treatment were significantly higher than those observed in the control. The results indicate that N application significantly stimulated soil enzyme activity, attributed to the increase in N concentration, which also enhanced P content in the soil. Together, the increased enzyme activities and soil nutrient availability promoted yield components and seed yield in *F. kirilowii*.

## ACKNOWLEDGEMENTS

The authors thank the experts for editing our article and the anonymous reviewers for their critical comments and suggestions to improve this article.

### Funding

This research was supported by the Qinghai innovation platform construction project (2023), and the China Agriculture Research System (CARS-34). The funders had no role in study design, data collection and analysis, decision to publish, or preparation of the manuscript.

### Grant Disclosures

The following grant information was disclosed by the authors:
Qinghai innovation platform construction project (2023).
China Agriculture Research System (CARS-34).

### Competing Interests

The authors declare that they have no competing interests.

## Author Contributions

- Wenbo Mi conceived and designed the experiments, performed the experiments, analyzed the data, authored or reviewed drafts of the article, and approved the final draft.
- Feng Luo performed the experiments, authored or reviewed drafts of the article, and approved the final draft.
- Wenhui Liu conceived and designed the experiments, authored or reviewed drafts of the article, and approved the final draft.
- Yan Qin analyzed the data, prepared figures and/or tables, and approved the final draft.
- Yongchao Zhang analyzed the data, prepared figures and/or tables, and approved the final draft.
- Kaiqiang Liu analyzed the data, authored or reviewed drafts of the article, and approved the final draft.
- Wen Li analyzed the data, prepared figures and/or tables, and approved the final draft.

## Data Availability

The raw data is available in the Supplemental File.

## Supplemental Information

Supplemental information for this article can be found online at http://dx.doi.org/10.7717/peerj.16791#supplemental-information.

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
