# Peer review of "Nitrogen addition enhances seed yield by improving soil enzyme activity and nutrients"

_PeerJ, doi:10.7717/peerj.16791_

## Round 0.1 · original submission · Major Revisions

Dear Dr. Mi,
Thank you for your submission to PeerJ.
It is my opinion as the Academic Editor for your article - Nitrogen addition enhances seed yield by improving soil enzyme activity and nutrients - that it requires a number of Major Revisions.
My suggested changes are reported in an annotated pdf file; reviewer comments are shown below and on your article 'Overview' screen.
Please address these changes and resubmit. Although not a hard deadline please try to submit your revision within the next 35 days.
Kind Regards,

Valeria Spagnuolo

**Language Note:** The review process has identified that the English language must be improved. PeerJ can provide language editing services - please contact us at copyediting@peerj.com for pricing (be sure to provide your manuscript number and title). Alternatively, you should make your own arrangements to improve the language quality and provide details in your response letter. – PeerJ Staff

Reviewer 1 ·

Basic reporting

Some nouns need to be written normatively

Experimental design

no comment

Validity of the findings

no comment

Additional comments

This study conducted an experiment over a two years period with N addition in forage seed field in the QTP, the results showed that the application of different concentrations of N fertilizers can stimulate soil enzyme activity, thereby accelerating nutrient conversion and thus increasing seed yield. The suggestions are as follows:
1. Line 82, there should be ‘between 2022 and 2023’
2. Precipitation data for 2022 and 2023 should be given. In fact, precipitation is an important factor affecting the primary productivity of alpine grassland.
3. Throughout the whole manuscript, PH should be pH
4. Line 235, NH4+ , NO3-
5. For the purposes of this study, data of 0-20 cm were enough.
6. The suitable amount of fertilizer may be more appropriate as a range, especially with the local precipitation.

Reviewer 2 ·

Basic reporting

English requires check. Too many acronyms and one is left confused
Though the work is good scientifically, paper required editing, restructuring of sentences and proofreading. I suggest adding tables and flowsheets to help reader understand how the manuscript flows.

Experimental design

Good. Scientifically good paper

Validity of the findings

Yes all data are robust and statistically sound and can be replicated and reproduced

Additional comments

Please rewrite using simple sentences and reduce acronyms

Reviewer 3 ·

Basic reporting

The manuscript sheds light on a pertinent topic: the impacts of nitrogen addition on soil enzyme activities, soil nutrients, and the seed yield of F. kirilowii in the Qinghai-Tibet Plateau. The findings have the potential to augment our understanding of forage seed production and ecological restoration within this region. However, there are areas that warrant improvement.

While the literature review is robust and relevant, it seems to rely on older publications. I recommend integrating more recent studies to ensure the review is current.
There are instances in the 'Materials and Methods' section where certain methodologies, such as Kjeldahl digestion, are mentioned without adequate elaboration. Though these might be well-known within the field, citing a pertinent reference would enhance clarity.

The manuscript would benefit from thorough proofreading. Some specific errors noted include:
Line 147, Page 8: "PH" should be corrected to "pH".
Line 234, Page 10 & Line 273-274, Page 11: Chemical ions (NH+4, NO2- and NO-3) are incorrectly represented. The correct forms are NH4+, NO2- and NO3-.
Line 39-40, Page 6: The sentence structure is convoluted. A possible revision could be: "Previous studies on Elymus nutans have shown that an optimal N addition rate of 250 kg·hm² can enhance seed yield."
Grammar mistakes in Line 41: Yuan et al. (2022)ís study that examined the impact of amounts of N addition ...
The abbreviation "CK" for the control group should be introduced in its full form for clarity.

Regarding Figure 6, there's a discrepancy between the text and the visual representation. The text mentions red lines indicating a negative relationship, yet no such lines are discernible in the figure. The authors should revisit this and rectify any inconsistencies.

Experimental design

The introduction efficiently sets the stage by providing relevant background. However, it would benefit from a more explicit delineation of the research question. The authors should pinpoint gaps and shortcomings in current research and clearly articulate the significance and objectives of this study in bridging these gaps.

The experimental design is rigorous and follows a randomized complete block design with seven nitrogen treatments and three replicates. However some methods are described with not enough detail. For example, the authors do not report how they ensured balanced covariates between groups. They also do not report any baseline characteristics of the soil or any tests for group equivalence on soil properties.

Validity of the findings

The results are robust, statistically sound, and controlled. The authors report the main findings of the data analysis in a logical and transparent manner. However, some results are not well explained or discussed in the text. For example, the authors do not explain why soil pH decreased with nitrogen addition, or why soil enzyme activities showed different responses to nitrogen addition. The authors should provide more interpretation and discussion of their results in relation to their research question and hypothesis, as well as to the existing literature on nitrogen addition and seed yield of F. kirilowii.

The discussion section offers valuable insights, but there's an opportunity for further elaboration. It would be beneficial if the authors could delve deeper into methods for determining the optimal N addition as informed by this study. Additionally, a more comprehensive summary of the main findings, coupled with an exploration of their implications and any limitations, would enrich the discourse. Suggestions for future research directions would also be a valuable addition.

Reviewer 4 ·

Basic reporting

The manuscript entitled " Nitrogen addition enhances seed yield by improving soil enzyme activity and nutrients" by Wenbo and colleagues investigates the impact of Nitrogen fertilization on soil enzyme activities, soil nutrients, and the seed yield of Festuca kirilowii in the Qinghai-Tibet Plateau. The study was conducted over a period of two years. The results show that the application of different concentrations of N stimulates soil enzyme activities, leading to increased yield components and overall seed yield. Providing practical suggestions for improving seed production of F. kirilowii in the Qinghai-Tibet Plateau. Overall, this manuscript is analyzed in a logic, organized, and statistically tested manner. However, there are a few points that need clarification or further details.

Experimental design

1. How are the three replicated plots arranged for each N concentration?
2. Are there any notable plant growth phenotype observed other than seed yield? Are there any photos taken?
3. During July and September, how does the rain affect the result since the rain contains nitrogen as well?

Validity of the findings

1. The R code is missing.
2. For all the charts, I suggest using boxplot showing all the datapoints for replicates.
3. The annual weather conditions (precipitation, temperature, evaporation, etc.) were provided, are the conditions in between 2021 and 2023 similar?

Additional comments

1. Please add space in front of all in-text citations and comma in after et al.. Example: line 30, "most plants (Amanullah et al., 2016)"
2. Line 39: Missing a verb between “could seeds”.
3. Line 78: I suggest changing “a two years” to “a two-year”.
4. Line 89: "S1.." should be "S1."
5. Line 96-97: please rephrase this sentence for an improved clarity.
6. Line 113: please add kit’s name.
7. Line 122: “Soil organic matter (OC)” should be (SOM)
8. Line 123: Please provide details of Kjeldahl digestion and perchloric acid digestion.
9. Figure 6: I suggest adding the full names for all the abbreviations of the enzymes in the legend will help. These are all black lines, where are red color lines?

---

## Round 0.2 · Minor Revisions

The authors have done extensive work to improve the substance and form of their MS, addressing all the main points raised by the reviewers and the editor. However, there are still some minor revisions. Therefore, I invite the authors to correct the relevant points according to the indications of reviewers 1 and 3; after this, the MS can be accepted for publication.

Reviewer 1 ·

Basic reporting

The standard writing of ammonium and nitrate ions still needs to be modified.

Experimental design

No comment

Validity of the findings

No comment

Additional comments

No comment

Reviewer 2 ·

Basic reporting

All comments have been addressed

Experimental design

N/A

Validity of the findings

N/A

Reviewer 3 ·

Basic reporting

The revised manuscript has commendably addressed the previous comments, reflecting the authors' dedication to improving their work. However, to ensure the manuscript reaches its full potential, I recommend conducting a final thorough proofreading.

For instance, in Line 120, there is a lingering typographical error: "nitrate reductase [NR)" should be corrected to "nitrate reductase (NR)." Additionally, in Lines 257 and 259, it is advisable to subscript the numbers in "NH4+" and "NO3-" and superscript "plus" and "minus" to align with established chemical notation standards.

Furthermore, I suggest the authors enhance the clarity of the figures by including denotations for the meaning of "a," "b," "c," and "d" in the figure captions for Figures 1-3. This adjustment could significantly improve the comprehensibility of the figures.

Experimental design

no comment

Validity of the findings

no comment

Reviewer 4 ·

Basic reporting

Based on revision of manuscript, authors successfully incorporated the point to point wise through the manuscript as suggested comments so that I would like to strongly recommend that acceptance of manuscript in your prestigious journal for publication.

Experimental design

No additional comments.

Validity of the findings

No additional comments.

---

## Round 0.3 · accepted · Accept

I personally checked that the authors have addressed all concerns and comments raised by the reviewers; therefore, the manuscript is now ready for publication.